# Semi-supervised Ordinal Regression via Cumulative Link Models for Predicting In-Hospital Length-of-Stay

Alexander A. Lobo [1]  Preetish Rath [1]  Michael C. Hughes [1]

## Abstract

Length-of-stay prediction has been widely studied as a classification task: will this patient stay 0-3 days, 3-7 days, or more than 7 days? Yet previous approaches neglect the natural ordering of these classes: standard multi-class classification treats classes as unordered, while methods that build separate binary classifiers for each class struggle to enforce coherent probabilistic predictions across classes. Instead, we suggest that cumulative link models (CLMs), an ordinal approach long-known in statistics, are a naturally coherent approach well-suited to predicting length-of-stay. We describe how CLMs can be integrated as an output layer into any training pipeline based on automatic differentiation.[2] We show that CLM output layers yield competitive predictions over binary classifier alternatives when paired with either neural net or hidden Markov model representations of patient vital sign history, all while requiring fewer parameters. Further experiments show promise in a semi-supervised setting, where only some patients have observed outcomes.

## 1. Introduction

As the world's population ages and impacts of diseases become more varied, the length of stay (LOS) of a typical patient in critical care may increase and put a significant burden on hospitals (Akinosoglou et al., 2023; Daghistani, 2019). There are two main reasons why predicting patient LOS is a useful task for intensive care units (ICUs) (Almashrafi, 2016; Clarke & Rosen, 2001). First, *resource allocation*: Understanding how long patients will stay in the ICU can help hospitals plan for the necessary staffing, bed

availability, and medical resources, thereby reducing costs. Second, *patient care planning:* Physicians and care teams can formulate plans for patients predicted to have longer LOS, such as monitoring to prevent hospital-acquired infections or other health complications.

In many cases, the utility derived from LOS prediction comes not from a precise numerical prediction (e.g. distinguishing between a stay of 8.5 and 9.1 days), but instead from knowing a coarse class (e.g. patient will likely stay more than 7 days). Coarse classes are often sufficient to determine a stakeholder decision or action. Hence, LOS prediction is often (though not always) framed as a classification problem, usually with multiple possible classes.

As an example of the state-of-the-art, a recent LOS classifier by Cai et al. (2022) handles multi-class prediction by coordinating separate binary classifiers: one decides if the stay will exceed 3 days, another if LOS will exceed 7 days, etc. While these models can answer binary questions in isolation, overall coherence matters. Separately trained models might assign higher probability to a $>7$ day stay than a $>3$ day stay, which is logically incoherent since the latter includes the former. Cai et al. (2022) use an extra loss to try to avoid such incoherence, but lack formal guarantees.

Instead, we suggest a family of statistical likelihoods designed for ordinal data, known as *cumulative link models* or CLMs (Christensen, 2022). In addition to naturally handling the ordered nature of LOS classes in coherent fashion, CLMs are *less complicated*: they need fewer parameters (just one weight vector instead of one per class) and no extra loss functions to ensure coherence. While CLMs have long been known in the statistics literature (Aitchison & Silvey, 1957; McCullagh, 1980; Chu & Ghahramani, 2005), this approach appears under-studied for modern LOS prediction. A recent survey of LOS prediction models (Bacchi et al., 2022) identified 21 high-quality studies. Of these, 7 framed the task as multi-class classification (others pursued regression or a single binary task). Exactly zero used cumulative link methods. One barrier might be that the suitability of CLMs for modern gradient-based learning is not obvious, especially to non-experts.

We hope in this paper to argue that when classification is

---

[1]Department of Computer Science, Tufts University, Medford, MA, USA. Correspondence to: Michael C. Hughes <michael.hughes@tufts.edu>.

*Workshop on Interpretable ML in Healthcare at International Conference on Machine Learning (ICML)*, Honolulu, Hawaii, USA. 2023. Copyright 2023 by the author(s).

[2]Code URL: github.com/tufts-ml/cumulative-link-models

the right framing for an applied LOS task, CLMs can be revitalized as a reliable and interpretable model relevant to hospitals today. We develop a view of CLMs as an output layer of a neural network amenable to automatic differentiation and release open-source code (link on page 1). This work can be considered as a "missing manual" for a separately developed but insufficiently documented OrderedLogistic layer available in Tensorflow Probability (Tensorflow Developers, 2023).

Furthermore, CLMs may be incorporated into semi-supervised frameworks to address prediction tasks where only some hospitalized patients have associated outcome labels of interest. We particularly look at integrating CLMs with prediction-constrained hidden Markov models (PC-HMMs) (Hope et al., 2021), which our team has previously demonstrated to be effective for semi-supervised binary classification of vital sign time series (Rath et al., 2022), but not ordinal tasks. Although we don't envision a scenario where LOS labels are unavailable in practice, we use it to showcase the utility of PC-HMMs with a CLM output layer for predicting any clinical outcome with ordinal labels. Future work could apply such models to disease severity prediction tasks, where diagnostic labels may be naturally available only for a subset of all patients.

## 2. Background and Related Work

In this paper, we pursue LOS prediction as a supervised learning task. We observe a labeled dataset of $(x, y)$ pairs. Given observed measurements $x$ for a patient, we build a feature vector $\phi(x) \in \mathbb{R}^F$ using any suitable representation (neural network, etc.). Each class label $y \in \{1, 2, \dots C\}$ denotes which of $C$ *ordered* levels of LOS classes that patient belongs to. Our goal is to learn a probabilistic model that can predict $p(y = c|x)$ for each class $c$. Below, we survey the extensive body of work on LOS predictions, focusing on how $C > 2$ class labels are modeled.

**Unordered multi-class classifiers.** Many approaches pursue multi-class classification without regard to class ordering. For example, a softmax construction yields likelihood

$$p(y = c|x) \propto \exp(w_c^{\mathsf{T}}\phi(x)) \qquad (1)$$

This approach requires $C$ different weight vectors $w_c \in \mathbb{R}^F$, one for each class. Harutyunyan et al. (2019) and Song et al. (2018) take this direction for LOS prediction, minimizing multi-class cross-entropy (an equivalent objective).

**Coordinated binary classifiers.** Other works try to coordinate multiple binary classifiers to obtain useful predictions over the $C$ classes, following Li & Lin (2006)'s so-called extended binary representation. For examples of uncritical application of multiple binary classifiers to LOS, see Daghistani (2019). Cai et al. (2022) proposed an output

layer that performs $C$ separate binary classifications, using a monotonicity constraint penalty to maintain the ordinal constancy. This implementation of separate binary classifiers is not able to constrain a monotonically decreasing probability prediction from shorter to longer LOS classes. That is, it is logically inconsistent to have $p(Y > 3|x)$ less than $p(Y > 7|x)$. Cao et al. (2020)'s CORAL (consistent rank logits) has some guarantees, but performs worse in Cai et al.'s tests.

**Previous multi-class remedies.** Several works aim to use multi-class methods but account for class ordering by modifying loss functions to favor nearby classes. Diaz & Marathe (2019) present SORD, which modifies multi-class to have soft labels that penalize nearby classes less than others. SORD uses a linear exponential. DLDL (Gao et al., 2017) is similar, using a squared exponential for soft labels.

**LOS as regression.** Other approaches frame the LOS task as regression. Yet mean absolute errors typically cover multiple days: 2.2 days in Rocheteau et al. (2021) and 4.6 days in Baek et al. (2018). This limited resolution suggests coarser classification, like we pursue, can sometimes be a more suitable goal.

**LOS via HMMs.** Sotoodeh & Ho (2019) propose a two-stage process that first trains an unsupervised hidden Markov model to learn latent representations of the features $x$ only, then fits a LASSO model on this fixed representation to make predictions of $y$. Our PC-HMMs (Hope et al., 2021; Rath et al., 2022) employ supervision to inform the HMM directly using labels $y$ and handle missingness via exact marginalization rather than mean imputation.

## 3. Methods

### 3.1. Cumulative Link Models for Ordinal Regression

We wish to model the label $y \in \{1, \dots, C\}$ given an $F$-dimensional feature vector $\phi(x)$, by mapping features to a scalar location $g \in (-\infty, +\infty)$ on the real line via a generalized linear model $g(x) = \eta^{\mathsf{T}}\phi(x)$ with weights $\eta \in \mathbb{R}^F$. A *deterministic* model might then map this location to a class via a set of ordered thresholds $\theta_0 < \theta_1 < \dots < \theta_C$:

$$p(y = c|x) = \begin{cases} 1 & \text{if } \theta_{c-1} < g(x) \leq \theta_c \\ 0 & \text{otherwise} \end{cases} \qquad (2)$$

where we fix boundaries at $\theta_0 = -\infty$ and $\theta_C = +\infty$. But the above would be flat almost everywhere as a function of parameters $\eta, \theta$, making gradient-based learning difficult.

*Cumulative link models* assume a smoother likelihood:

$$p(y=c|x) = H\left(\frac{\theta_c - g(x)}{\sigma}\right) - H\left(\frac{\theta_{c-1} - g(x)}{\sigma}\right) \quad (3)$$

where $H$ is a cumulative distribution function (CDF) of a location-scale univariate distribution $\mathcal{H}$ over the reals, and

$\sigma > 0$ is a scale parameter. Because $H$ as a CDF is a smooth increasing function, according to Eq. (3), increasing the value of $g$ gradually across any threshold $\theta_c$ leads to gradual changes in output probability (see Fig. A.2), not the sudden change across boundaries of Eq. (2).

If $H$ is the Normal CDF, we recover the cumulative probit model (Aitchison & Silvey, 1957; Chu & Ghahramani, 2005). If $H$ is the Logistic CDF, we recover the cumulative logit model (McCullagh, 1980), also called "proportional odds". See Agresti (2012) for textbook coverage of such models. Christensen (2022) presents a recent unifying perspective, even suggesting other possible distributions for $\mathcal{H}$. For concreteness, we focus here on Logistic and Normal.

We can interpret our target likelihood in Eq. (3) as the marginal $p(y|x)$ of an expanded model $p(y, \delta|x)$ that first samples a latent zero-mean noise variable $\delta \sim \mathcal{H}(\text{loc}=0, \text{scale}=\sigma)$, then determines the class $y$ by thresholding a noise-perturbed location $g + \delta$, finding $c \in \{1, \ldots, C\}$ such that $\theta_{c-1} < g + \delta \leq \theta_c$.

### 3.2. Implementation as a Differentiable Layer

We can view the CLM likelihood in Eq. (3) as an "output layer" that can be composed on top of any suitable representation layer $\phi(x)$. For example, we could set $\phi(x)$ to the features of a recurrent neural network or (as described in the next section) a hidden Markov model framework that naturally handles both missing features and semi-supervision.

One issue is that our likelihood is over-parameterized. Multiple combinations of $\eta, \theta$, and $\sigma$ values can yield the same likelihood. For instance, given any valid setting of $\eta, \theta, \sigma$, if we pick any constant $k > 0$, we can construct another set of parameters $\eta' = k\eta, \theta' = k\theta$, and $\sigma' = k\sigma$ that produce the same likelihood value for any input $x$, because $H(\frac{k\theta_c - kg}{k\sigma}) = H(\frac{\theta_c - g}{\sigma})$. To address this, we recommend fixing $\sigma = 1$, though any fixed value would work.

Another issue relates to the constraints of the threshold parameters. To be amenable to gradient descent, we need all parameters to be *unconstrained*. Given an unconstrained vector $\nu \in \mathbb{R}^{C-2}$, we construct thresholds iteratively:

$$\theta_1 = 0, \ \theta_2 = \theta_1 + r(\nu_1), \ldots, \ \theta_{C-1} = \theta_{C-2} + r(\nu_{C-2}) \quad (4)$$

where $r$ is the softplus function that maps a real scalar to a positive scalar. We can then write our output layer as $\text{CLMLAYER}(\phi(x); \eta, \nu) : \mathbb{R}^F \to \Delta^C$. This layer has two free parameters: one $F$-dimensional weight vector $\eta$ and a $(C-2)$-dimensional threshold-determining vector $\nu$. For any $C > 2$, this is always more compact than the earlier multi-class or separate binary approaches, which require $\mathcal{O}(CF)$ weights given the same representation $\phi(x)$. We observe in our experiments that freezing $\theta$ to reasonably spaced defaults (rather than learning thresholds via a free

parameter $\nu$) is often sufficient for good performance.

### 3.3. Time-Series Representation Learning

We view LOS prediction as a time-series classification task. Raw features $x = x_1, \ldots x_T$ can represent a multivariate time series of vital signs or other health signals (labs, medications, etc.) observed at hourly intervals for $T$ hours. In real hospitals, each $D$-dimensional vector $x_t \in \mathbb{R}^D$ may have arbitrary missingness, as some vitals/labs are measured only occasionally. This calls for representations $\phi(x)$ that are well-suited for time-series data with high potential missingness.

In the case of deep neural networks, we can set $\phi(x)$ to be the hidden state vector of a Gated Recurrent Unit given all $T$ timesteps in $x$. Missing features are either imputed via a heuristic like forward-filling (Harutyunyan et al., 2019) or imputed by an end-to-end-trained missingness-aware supervised model like BRITS (Cao et al., 2018).

Alternatively, we consider the *prediction-constrained hidden Markov model* (Hope et al., 2021). Rath et al. (2022) suggest this is a natural approach for prediction tasks with health record time series because it handles missingness elegantly. Given a $K$-state HMM with known transition and emission parameters, we can featurize a sequence by setting $\phi(x)$ to the average belief vector over all timesteps (see App. B). $\phi(x)$ is computable via dynamic programming; parameter learning is possible via automatic differentiation.

**SSL extensions.** Another advantage of the PC-HMM is the ability to naturally learn from a small labeled set and a large unlabeled set, taking advantage of the HMM's nature as a generative model to improve representations given unlabeled data. See Appendix for details.

## 4. Experimentation

**MIMIC-IV and eICU analysis.** We study data from MIMIC-IV (Johnson et al., 2016) to predict length of stay for 52,354 de-identified ICU patient-stays from one hospital. We pre-process the dataset to contain 38 time varying measurements of labs and vitals and 2 demographics (see App. D for full feature list, LOS distribution, train/valid/test splits, and missing data information). Building on prior works (Wang et al., 2020; Gong et al., 2017; Nestor et al., 2018; Rajkomar, 2018; Zebin et al., 2019), we predict LOS into 4 ordinal categories (<3 days, 3-7 days, 7-11 days, and >11 days) using only hourly measurements of the first 24 hours of patient stays. We remove patient stays less than 30 hours to avoid label leakage. To directly compare the ordinal regression models to binary classification models, we aggregate the predicted probabilities of each ordinal category and report performance for predicting binary outcomes (LOS >3 days, >7 days or >11 days). We focus

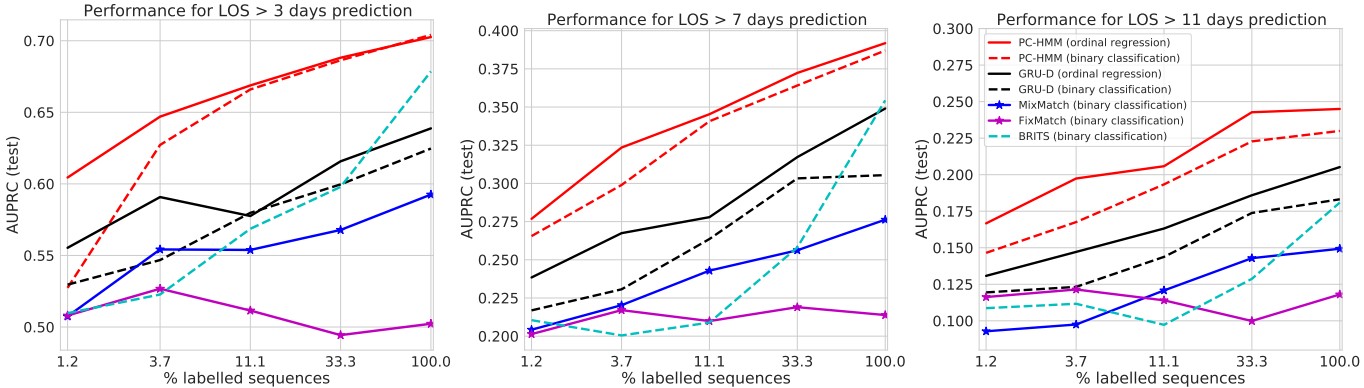

Figure 1: AUPRC (higher is better) versus amount of labeled data for length-of-stay prediction at 3 ordinal labels ($> 3$, $> 7$, and $> 11$ days). SSL methods (PC-HMM, MixMatch, and FixMatch) learn from both labeled and unlabeled data. GRU-D (Che et al., 2018) and BRITS (Cao et al., 2018) use the labeled set only. Binary classification models require 3 separate models (1 for each label), whereas ordinal regression using CLMs require a single model to predict all 3 labels.

on binary outcomes to show that CLMs can compete even when metrics favor binary not ordinal models. We do plan to expand our analysis to non-binary performance metrics in future work. We conduct similar LOS predictions on the eICU dataset (Pollard et al., 2018) (details in App. E). We predict LOS for slightly different ordinal categories ($<3$ days, 3-5 days, 5-7 days, and $>7$ days) to ensure a similar class-imbalance as MIMIC-IV (see Tables D.1 and E.1).

First, Table 1 compares separate binary models with $CF$ weights and our proposed ordinal model with $F$ weights, using both Normal and Logistic links. We do not expect our models to outperform these over-parameterized binary alternatives. Instead, we aim to show that our ordinal model can compete with the binary models while being significantly more efficient and coherent.

Next, Figures 1 and C.1 plots AUPRC as a function of

Table 1: Performance and coherency comparison between binary classification and ordinal regression results on MIMIC-IV with PC-HMM using different link functions (fully supervised). We label a prediction to be incoherent if the monotonicity constraint is violated (for example if $p(\text{LOS} > 3 \ days) < p(\text{LOS} > 11 \ days)$)

|  | BINARY CLASSIFICATION | | ORDINAL REGRESSION VIA CLM | | | |
|---|---|---|---|---|---|---|
| LINK $f$ | LOGIT | | LOGIT | | PROBIT | |
| MODELS PARAMETERS | 3 1338 | | 1 446 | | 1 446 | |
| INCOHERENCE[†] | 876 (8.3%) | | 0 (0%) | | 0 (0%) | |
| TASK | AUPRC | AUROC | AUPRC | AUROC | AUPRC | AUROC |
| LOS $> 3$ | 0.704 | 0.718 | 0.703 | 0.726 | 0.700 | 0.725 |
| LOS $> 7$ | 0.387 | 0.766 | 0.392 | 0.771 | 0.389 | 0.767 |
| LOS $> 11$ | 0.230 | 0.763 | 0.245 | 0.786 | 0.243 | 0.783 |

[†]Refers to portion of stays in test-set predictions that are incoherent.

available labeled data in an SSL setting for MIMIC-IV and eICU respectively. Two major takeaways are apparent. First, ordinal regression CLMs (solid lines) outperform binary classifiers (dashed) regardless of whether HMM or neural representations are used. Second, compared to all other baselines, the ordinal regression PC-HMM not only achieves the best performance but also does so with the fewest trainable parameters and lowest training time (see Table F.1).

**Interpretability.** We illustrate the interpretability of the PC-HMM in Figure 2. It showcases the learned parameters of the emissions distributions and plots the average beliefs of patients in the 'risk' state across various length of stay categories. The 'risk' state refer to the states with the highest predictor weight $\eta$ (see equation 8). Notably, the PC-HMM effectively identifies patients at risk by considering high creatinine and low albumin levels. Additionally, as the true length of stay of increases, the fraction of patients with high average beliefs in the risk states also increases. This demonstrates that the PC-HMM accurately associates longer stays with abnormal physiological measurements.

## 5. Conclusion

In order to address the growing need to predict reliable LOS outcomes from patient data, we have presented a model that not only outperforms alternatives but also does so with minimal parameters. By taking advantage of the smooth likelihood for ordinal data, we show how the cumulative link framework may be incorporated into any model trainable with gradient descent. When incorporated into the PC-HMM, we observe how generative and discriminative goals are balanced to effectively predict LOS in both fully- and semi-supervised cases. Unlike other candidates for ordinal regression, this established framework maintains a

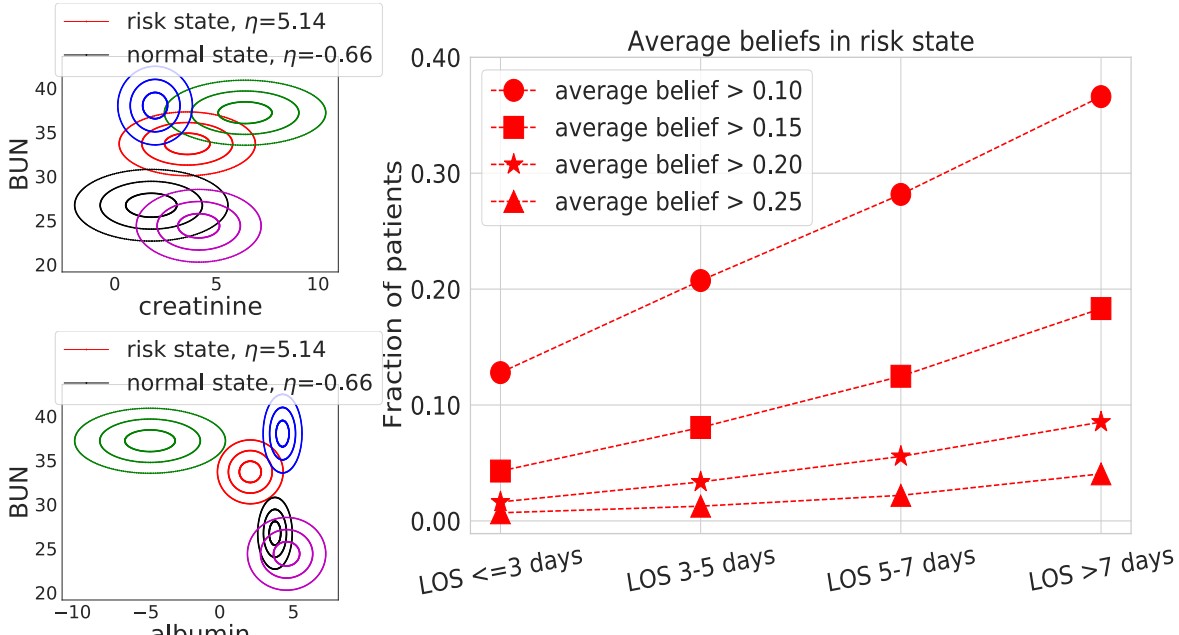

Figure 2: **Left** : Our PC-HMM model reveals five states, representing different risk levels. The red state indicates the 'risk state' associated with high BUN and low albumin and high creatinine. **Right**: We show the average beliefs, averaged over time, in the 'risk state'. Notably, as the LOS increases, the average beliefs in the 'risk state' increases.

monotonically increasing probability for subsequent ordinal classes that are grouped together. This, along with their loss convexity, make CLMs an ideal candidate to address the novel problem of predicting hospital stays. Promising results on patient time-series datasets exhibit the utility of this model. We plan to explore further use cases such as characterizing disease severity with medical data.

## Acknowledgments

We gratefully acknowledge computing hardware support from the U.S. National Science Foundation under grant NSF OAC-2018149.

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

# A. Cumulative Link Models

## A.1. Illustration of Likelihood

Below we observe the effects of the CLM likelihood by adjusting the choice of distribution $\mathcal{H}$ used. We notice that the Gaussian model (probit link function) provides more flexibility than the logistic model (logit link function) to adjust confidence of classification near cutpoints. When $\sigma = 1.0$, we observe a comparable likelihood function to that using the logistic distribution.

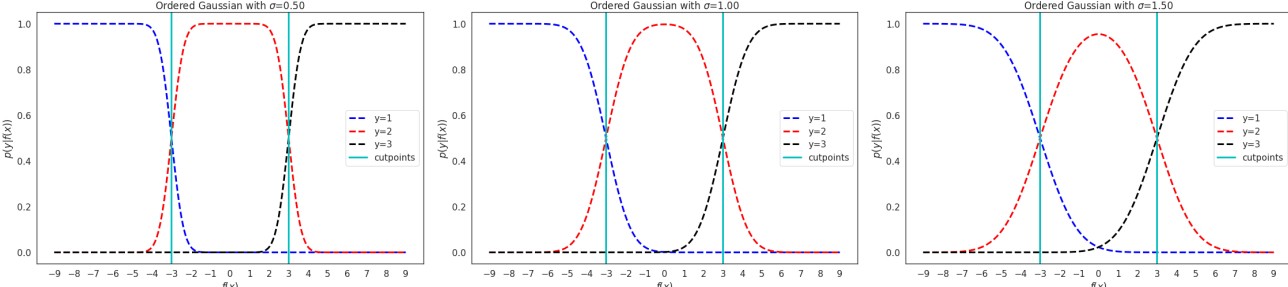

Figure A.1: Visualization of likelihood function for ordinal regression with 3 classes using the Gaussian distribution at various $\sigma$ values to simulate latent function noise

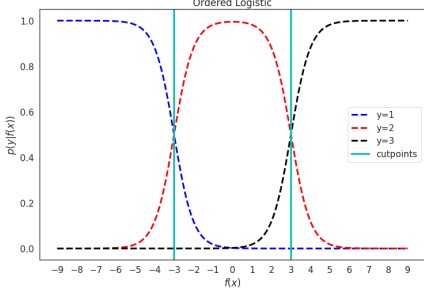

Figure A.2: Visualization of likelihood function for ordinal regression with 3 classes using the logistic distribution to simulate latent function noise

## A.2. Demo: CLMs for Ordinal Prediction from fixed features

Here we discuss the experimentation of our implemented ordinal regression model on non-timeseries data, which can also be thought of as a time series dataset with a single time step. The ordinal regression model was implemented in Python using the NumPy, SciPy, and Autograd libraries. First, a class is instantiated for a new model. Then, the model is fit to the training dataset by invoking a class method, which uses the equations listed in the main body to compute a negative log-likelihood for the observed data given some randomly chosen initial values for each of the model's parameters $\eta$ and $\theta$. The Autograd library is then used to compute a gradient on the loss function, which is then used with SciPy's optimize capabilities to find the values for the parameters that minimize the loss function.

It should be noted that some constraints were placed to ensure stable and logical training occurred. First, we consider that the ordered constraint of the ordinal regression can only be maintained with the cutpoints $\theta$ incrementally increasing. To avoid the gradient descent algorithm from violating this order, positive padding variables $\Lambda = [\Lambda_2, \Lambda_3, \dots \Lambda_{r-1}] \in \mathbb{R}^+$ were used in lieu of cutpoints such that any subsequent cutpoint can be computed from the first and sum of the appropriate set of padding variables: $\theta_j = \theta_1 + \sum_{\iota=2}^{j} \Lambda_\iota$. Now, rather than $\eta$, $\theta$, and $\sigma$ we have $\eta$, $\theta_1$, $\Lambda$, and $\sigma$ as our four sets of variables. The softmax activation function was used to ensure that padding variables are always positive. Second, to avoid unnecessarily increasing the moduli of the latent feature weight values and the padding variables to achieve diminishing

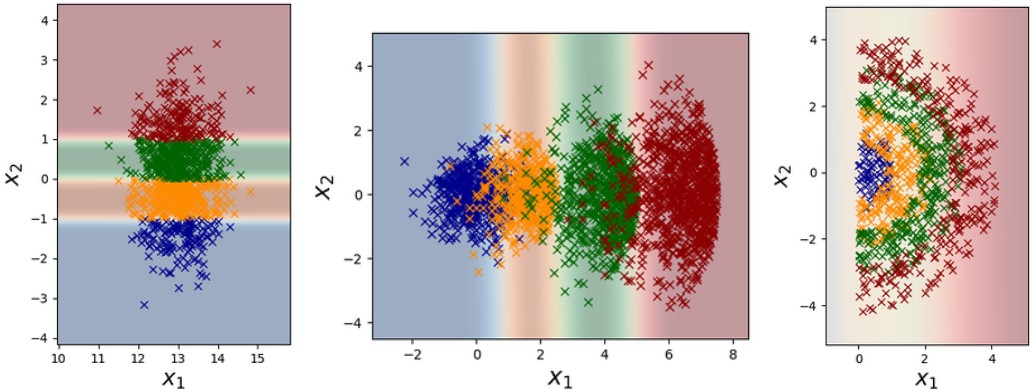

Figure A.3: Illustration of decision boundaries found by ordinal regression model on toy data. Colored markers represent ordinal labels in the following order: blue (y=0), yellow (y=1), green (y=2), red (y=3). **Left**: Appropriate boundaries are found for linearly separable data in the second feature dimension. **Middle**: More uncertainty is added to the decision boundaries with the overlap of labels. **Right**: Example where model is not able to learn non-linear curvature of the separating boundaries given the linear nature of the latent function.

returns in the loss, thereby over-fitting the model to the training data, a complexity penalty was added on the latent function weights.

The model was tested with various 2-D feature generated datasets. However, in preliminary testing, the optimizer was not able to compute a valid gradient for the loss function, and NaN values were returned for each of the variables. Upon further investigation, it was discovered that the optimal $z = \frac{\theta_c - \eta^{\mathrm{T}}\phi(x)}{\sigma}$ term(s) needed to compute the CDF for the CLM likelihood could be achieved with multiple different combinations of $\theta$, $\eta$, and $\sigma$ values, which led us to believe that the model was over-parameterized learning $\eta$, $\theta_1$, $\Lambda$, and $\sigma$. This issue was addressed by constraining $\sigma = 1$.

Although the results were promising, efforts to improve the model by performing all operations in log space to avoid underflow and overflow computational issues failed with the NumPy operation restrictions in the Autograd library. However, these improvements were addressed by implementing the model with TensorFlow Probability.

### A.3. Demo: CLMs for Ordinal Prediction from Time Series

The following toy example is meant to showcase the utility of the ordinal PC-HMM model under predictable conditions. When applied to a time-series toy data example, the model is able to identify the four states that determine the correct ordinal class (see Figure A.4).

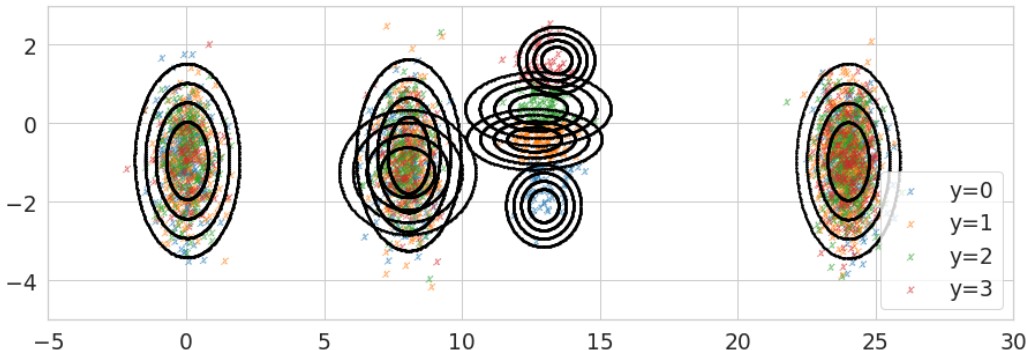

Figure A.4: Illustration of decision boundaries found by ordinal regression model on toy data. The PC-HMM correctly identifies the 4 clusters that are critical for prediction.

Once the PC-HMM learns the appropriate state sequences as indicated by the state beliefs, any observed sequence that has a

high probability of passing through the appropriate state is likely to have its respective ordinal class predicted correctly.

## B. Prediction Constrained Hidden Markov Models

A typical generative HMM assumes that $N$ observed sequences of data $X$ are generated from $N$ sequences of $K$ possible hidden discrete states $Z$, where each observed sequence $x_n$ and corresponding latent state sequence $z_n$ has $T_n$ time steps: $x_n = [x_{n1}, x_{n2} \ldots x_{nT_n}]$, $z_n = [z_{n1}, z_{n2} \ldots z_{nT_n}]$. The state sequences $z_n$ of an HMM are drawn from a Markov process, where each sequential state $z_{nt}$ has some probability of being observed conditioned on the previous state $z_{nt-1}$ (first order assumption), and the observed data for each time step in a sequence $x_{nt}$ is drawn from a distribution conditioned only on the respective state assignment. We assume Gaussian distributions are used to emit the observed data with a state-specific mean and covariance.

$$p(x_{nt}|z_{nt} = k, \epsilon_k = \{\mu_k, \Sigma_k\}) = \mathcal{N}(x_{nt}|\mu_k, \Sigma_k) \tag{5}$$

The HMM model can be comprehensively described as the joint probability of observing some data sequence and hidden state sequence:

$$p(x_n, z_n|\pi, \epsilon) = p(z_n|\pi)p(x_n|z_n, \epsilon) = \text{Cat}(z_{n1}|\pi_0) \underbrace{\prod_{t=2}^{T_n} \text{Cat}(z_{nt}|\pi_{z_{nt-1}})}_{p(z_n|\pi)} \underbrace{\prod_{t=1}^{T_n} \mathcal{N}(x_{nt}|\mu_{z_{nt}}, \Sigma_{z_{nt}})}_{p(x_n|z_n, \epsilon)} \tag{6}$$

where $\pi_0$ is the initial state distribution, and $\pi = \{\pi_k\}_{k=0}^{K}$ describes the outgoing transition probabilities from state $k$.

The posterior marginal probabilities (state beliefs) is the likelihood that a particular set of latent states generated the observed data. State beliefs may be averaged across timestamps and used as features to classify the data. In the binary classification case, the beliefs are multiplied by some regression coefficients $\eta$ and the logistic function is used as the appropriate link function:

$$\bar{b}(x_n, \pi, \epsilon) \triangleq \frac{1}{T_n} \sum_{t=1}^{T_n} b_t(x_n, \pi, \epsilon) \tag{7}$$

$$\hat{y}_n \triangleq \hat{y}(x_n, \pi, \epsilon, \eta) = \sigma(\eta^\top \bar{b}(x_n, \pi, \epsilon)) \tag{8}$$

The forward-backward algorithm can be used to compute the posterior marginal probabilities, or beliefs, for the latent states $z_n$, given the observed data $x_n$ (Rabiner, 1989). We can denote these probabilities by

$$b_{tk}(x_n, \pi, \epsilon) \triangleq p(z_{nt} = k|x_n, \pi, \epsilon) \tag{9}$$

Note that each of these beliefs are a deterministic function of $x_n$ with computational cost $\mathcal{O}(T_n K^2)$.

The probabilities $b_{tk}(x_n, \pi, \epsilon)$ take into account the full sequence $x_n$, including future timesteps $x_{nt'}$, where $t' > t$. In some applications, predictions must be made only on the data up until time $t$. These beliefs, defined as

$$\overrightarrow{b_{tk}}(x_n, \pi, \epsilon) \triangleq p(z_{nt} = k|x_{n1}, \ldots, x_{nt}, \pi, \epsilon) \tag{10}$$

are computed by the forward pass of belief propagation.

Hope et al. (2021) proposed a framework for training hidden Markov models while balancing generative and discriminative goals. The framework uses an HMM model to predict labels from beliefs while under a prediction constraint. Under this model, the generative component and the discriminative component, defined by separate parameters, are learnt jointly using a gradient descent algorithm. Previously, Rath et al. (2022) demonstrated how the PC-HMM model can be used for effective binary classification, despite feature and label missingness usually prevalent and persistent in clinical data. They reported precision-recall performance competitive with complicated models, and with many fewer parameters, for predicting mortality. This previous work exhibited how the PC-HMM's generative properties can be used to handle feature missingness and small labeled sets while maintaining effective prediction. Although it was argued that any prediction constraint may be used, so long as its loss function is differentiable, the previous work by Rath et al. (2022) only explored binary classification. Here, we extended the application of the PC-HMM to ordinal regression, specifically for predicting hospital length of stay (LOS).

## C. eICU Length of Stay Prediction Results

We pursue the task of length of stay prediction for stays $> 3$, $> 5$ and $> 7$ days on eICU with a much larger cohort of admissions (200,000). Again the PC-HMM and GRU-D with the ordinal regression loss outperform the respective models trained for binary classification

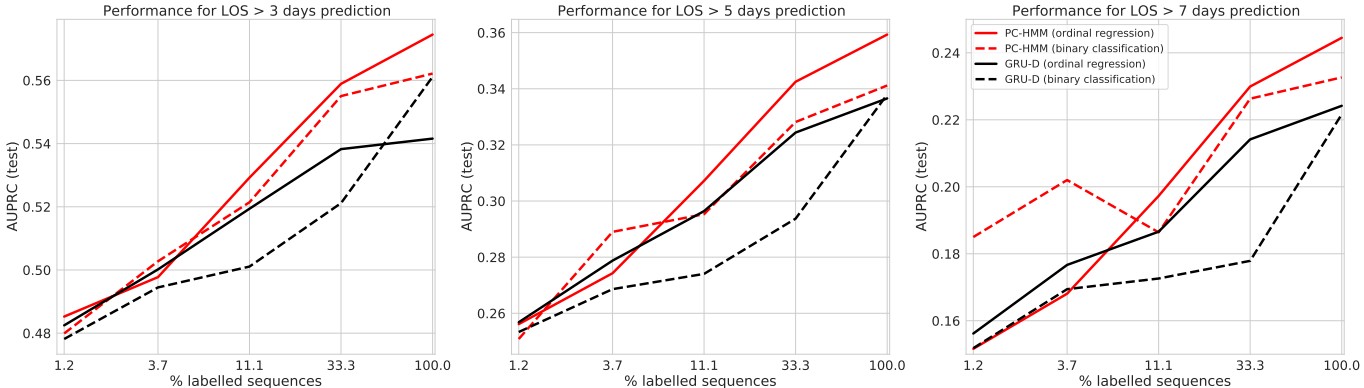

Figure C.1: AUPRC (higher is better) versus amount of labeled data for length of stay prediction at 3 ordinal labels ($> 3$, $> 5$, and $> 7$ days) on e-ICU. X-axis: Percentage of all training sequences available with labels (SSL methods treat remaining sequences as unlabeled; other methods discard them). Y-axis: Area under precision-recall curve (AUPRC, higher is better). SSL methods, including our PC-HMM as well as MixMatch and FixMatch, learn from both labeled and unlabeled data. GRU-D, BRITS, and Random Forest use the labeled set only. The PC-HMM matches or beats the other models across all tested labeled set sizes, despite needing fewer parameters and only $1/10^{th}$ of the training time as the other models. Additionally the models trained with the ordinal loss objective outperform the models trained with binary cross entropy. Note that for binary classification, 3 separate models are trained for each ordinal label, whereas a single cumulative link model is trained for ordinal regression to predict all 3 ordinal labels.

# D. MIMIC-IV Dataset Description

The MIMIC-IV data set utilized in this work contains the first 24 hours of 38 time-varying measurements and 3 demographics. We only include stays lasting more than 30 hours to ensure no label leakage (Wang et al., 2020). The length of stay distribution is shown below.

Table D.1: MIMIC-IV Dataset Split and Statistics

| Dataset | Split | Stays | Patients | Fraction of Stays | | |
| --- | --- | --- | --- | --- | --- | --- |
| | | | | LOS $\geq$ 3 days | LOS $\geq$ 7 days | LOS $\geq$ 11 days |
| MIMIC-IV | Train | 33,506 | 27,084 | 0.46 | 0.16 | 0.08 |
| | Valid | 8,377 | 7,821 | 0.46 | 0.16 | 0.08 |
| | Test | 10,471 | 9,673 | 0.47 | 0.17 | 0.08 |

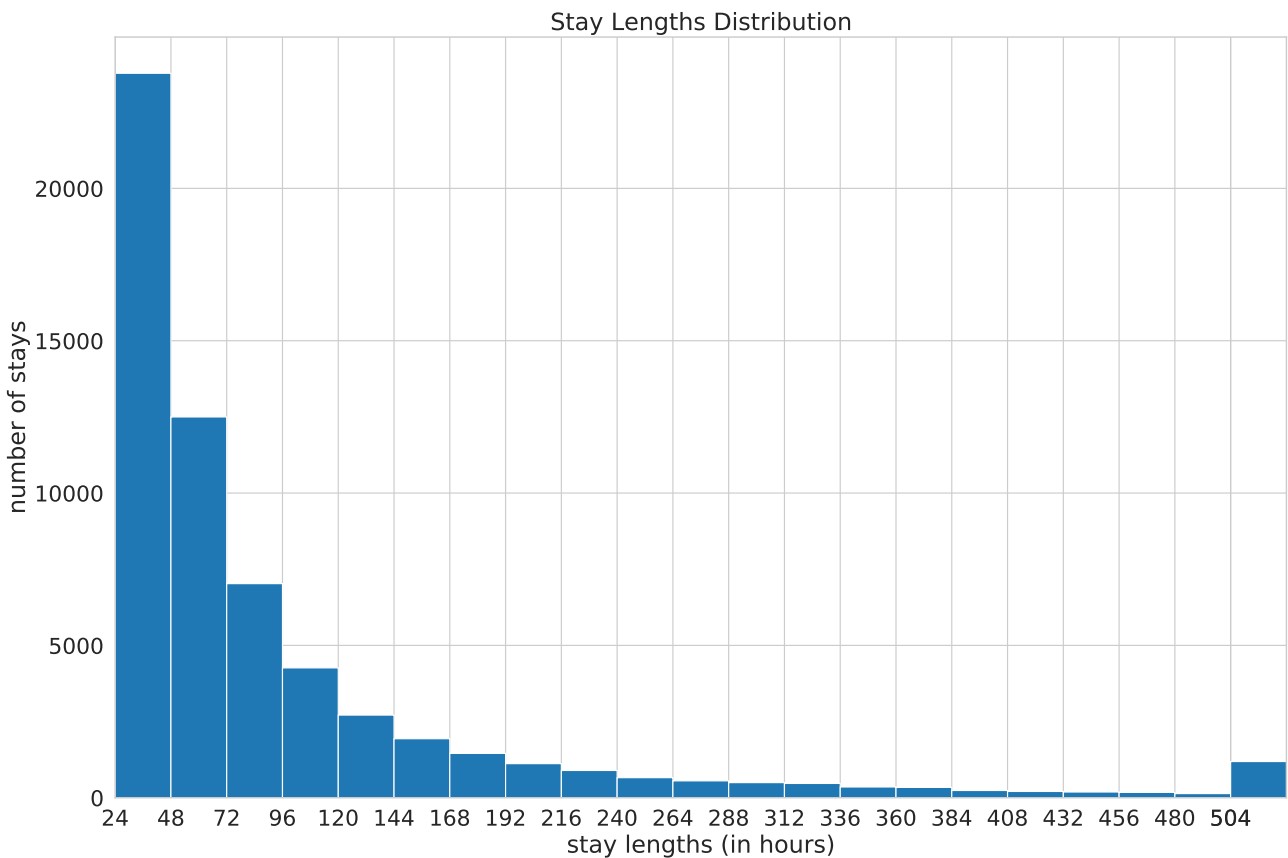

Figure D.2: Distribution of LOS in ICU obtained from MIMIC-IV

Table D.3: MIMIC-IV features

|  | 10% | median | 90% | missing_rate |
|---|---|---|---|---|
| Heart Rate | 63.00 | 83.00 | 109.00 | 0.00 |
| Respiratory Rate | 13.00 | 18.00 | 26.00 | 0.00 |
| O2 saturation pulseoxymetry | 93.00 | 97.00 | 100.00 | 0.00 |
| Non Invasive Blood Pressure systolic | 93.00 | 116.00 | 148.00 | 0.08 |
| Non Invasive Blood Pressure diastolic | 47.00 | 63.00 | 85.00 | 0.08 |
| Temperature Fahrenheit | 36.22 | 36.83 | 37.61 | 0.07 |
| Height (cm) | 155.00 | 170.00 | 183.00 | 0.98 |
| Respiratory Rate (Total) | 14.00 | 18.00 | 27.00 | 0.62 |
| Potassium (serum) | 3.40 | 4.10 | 5.10 | 0.04 |
| Sodium (serum) | 132.00 | 138.00 | 144.00 | 0.04 |
| Chloride (serum) | 96.00 | 104.00 | 112.00 | 0.04 |
| Hematocrit (serum) | 23.60 | 30.30 | 38.90 | 0.05 |
| Hemoglobin | 7.70 | 10.10 | 13.10 | 0.05 |
| Creatinine (serum) | 0.60 | 1.00 | 3.10 | 0.04 |
| Glucose (serum) | 88.00 | 126.00 | 222.00 | 0.05 |
| Magnesium | 1.60 | 2.00 | 2.60 | 0.08 |
| Phosphorous | 2.20 | 3.50 | 5.50 | 0.11 |
| Platelet Count | 79.00 | 175.00 | 322.00 | 0.05 |
| Glucose (whole blood) | 97.00 | 136.00 | 200.00 | 0.77 |
| Daily Weight | 58.40 | 82.30 | 114.00 | 0.66 |
| Absolute Neutrophil Count | 4.55 | 9.30 | 20.54 | 1.00 |
| Prothrombin time | 11.60 | 14.20 | 23.60 | 0.21 |
| Fibrinogen | 127.00 | 216.00 | 462.00 | 0.80 |
| PH (Arterial) | 7.25 | 7.37 | 7.46 | 0.60 |
| PH (Venous) | 7.23 | 7.36 | 7.45 | 0.78 |
| HCO3 (serum) | 17.00 | 23.00 | 28.00 | 0.05 |
| Arterial O2 pressure | 75.00 | 132.00 | 334.00 | 0.60 |
| Arterial CO2 Pressure | 31.00 | 40.00 | 52.00 | 0.60 |
| Lactic Acid | 1.00 | 2.00 | 5.60 | 0.47 |
| Albumin | 2.20 | 3.10 | 3.90 | 0.75 |
| Calcium non-ionized | 7.30 | 8.30 | 9.30 | 0.12 |
| C Reactive Protein (CRP) | 2.00 | 40.70 | 213.40 | 0.98 |
| ALT | 11.00 | 33.00 | 395.00 | 0.59 |
| AST | 17.00 | 48.00 | 568.00 | 0.59 |
| Direct Bilirubin | 0.30 | 1.60 | 7.80 | 0.97 |
| Total Bilirubin | 0.30 | 0.80 | 5.10 | 0.60 |
| Troponin-T | 0.01 | 0.07 | 1.60 | 0.71 |
| Venous CO2 Pressure | 30.00 | 42.00 | 63.00 | 0.82 |
| Age | 40.00 | 65.00 | 84.00 | 0.00 |
| is_gender_male | 0.00 | 1.00 | 1.00 | 0.00 |
| is_gender_unknown | 0.00 | 0.00 | 0.00 | 0.00 |

## E. eICU Dataset Description

The eICU data set utilized in this work contains the first 24 hours of 35 time-varying measurements and 3 demographics. We only include stays lasting more than 30 hours to ensure no label leakage (Wang et al., 2020). The length of stay distribution is shown below.

Table E.1: eICU Dataset Split and Statistics

| Dataset | Split | Stays | Fraction of Stays | | |
|---------|-------|-------|-------------------|-----------------|-----------------|
| | | | LOS $\geq$ 3 days | LOS $\geq$ 5 days | LOS $\geq$ 7 days |
| eICU | Train | 71,602 | 0.43 | 0.22 | 0.13 |
| | Valid | 17,901 | 0.43 | 0.21 | 0.13 |
| | Test | 22,376 | 0.43 | 0.22 | 0.13 |

Table E.3: eICU features

| Lab Test | 10% | Median | 90% | Missing Rate |
|----------|-----|--------|-----|--------------|
| ALT (SGPT) | 12.00 | 29.00 | 178.00 | 0.56 |
| AST (SGOT) | 14.00 | 35.00 | 268.00 | 0.55 |
| BUN | 9.00 | 21.00 | 59.00 | 0.07 |
| Hct | 23.40 | 31.30 | 40.90 | 0.08 |
| Hgb | 7.70 | 10.30 | 13.60 | 0.09 |
| MCH | 26.70 | 30.00 | 32.70 | 0.16 |
| PT | 11.60 | 15.60 | 27.00 | 0.61 |
| RBC | 2.63 | 3.57 | 4.62 | 0.10 |
| WBC x 1000 | 5.70 | 11.00 | 20.90 | 0.09 |
| albumin | 1.90 | 2.80 | 3.70 | 0.52 |
| anion gap | 6.00 | 11.00 | 18.00 | 0.26 |
| calcium | 7.20 | 8.20 | 9.20 | 0.09 |
| chloride | 96.00 | 105.00 | 113.00 | 0.07 |
| creatinine | 0.59 | 1.08 | 3.38 | 0.06 |
| glucose | 90.00 | 132.00 | 235.00 | 0.07 |
| platelets x 1000 | 87.00 | 181.00 | 313.00 | 0.10 |
| potassium | 3.30 | 4.00 | 5.00 | 0.05 |
| sodium | 132.00 | 139.00 | 145.00 | 0.06 |
| total bilirubin | 0.30 | 0.70 | 2.50 | 0.56 |
| total protein | 4.60 | 5.80 | 7.00 | 0.55 |
| uric acid | 2.70 | 6.10 | 11.90 | 0.98 |
| magnesium | 1.50 | 1.90 | 2.50 | 0.42 |
| bedside glucose | 91.00 | 137.00 | 239.00 | 0.37 |
| lactate | 0.80 | 1.90 | 5.70 | 0.71 |
| HCO3 | 16.20 | 22.90 | 31.00 | 0.62 |
| pH | 7.23 | 7.36 | 7.47 | 0.60 |
| FiO2 | 21.00 | 45.00 | 100.00 | 0.58 |
| Total CO2 | 17.20 | 24.00 | 35.30 | 0.85 |
| fibrinogen | 144.00 | 277.00 | 517.00 | 0.93 |
| CRP | 1.07 | 11.50 | 543.00 | 0.98 |
| phosphate | 1.90 | 3.30 | 5.50 | 0.62 |
| direct bilirubin | 0.10 | 0.30 | 2.40 | 0.91 |
| troponin - T | 0.02 | 0.14 | 1.96 | 0.96 |
| age | 41.00 | 65.00 | 82.00 | 0.03 |
| gender_is_male | 0.00 | 0.00 | 1.00 | 0.00 |

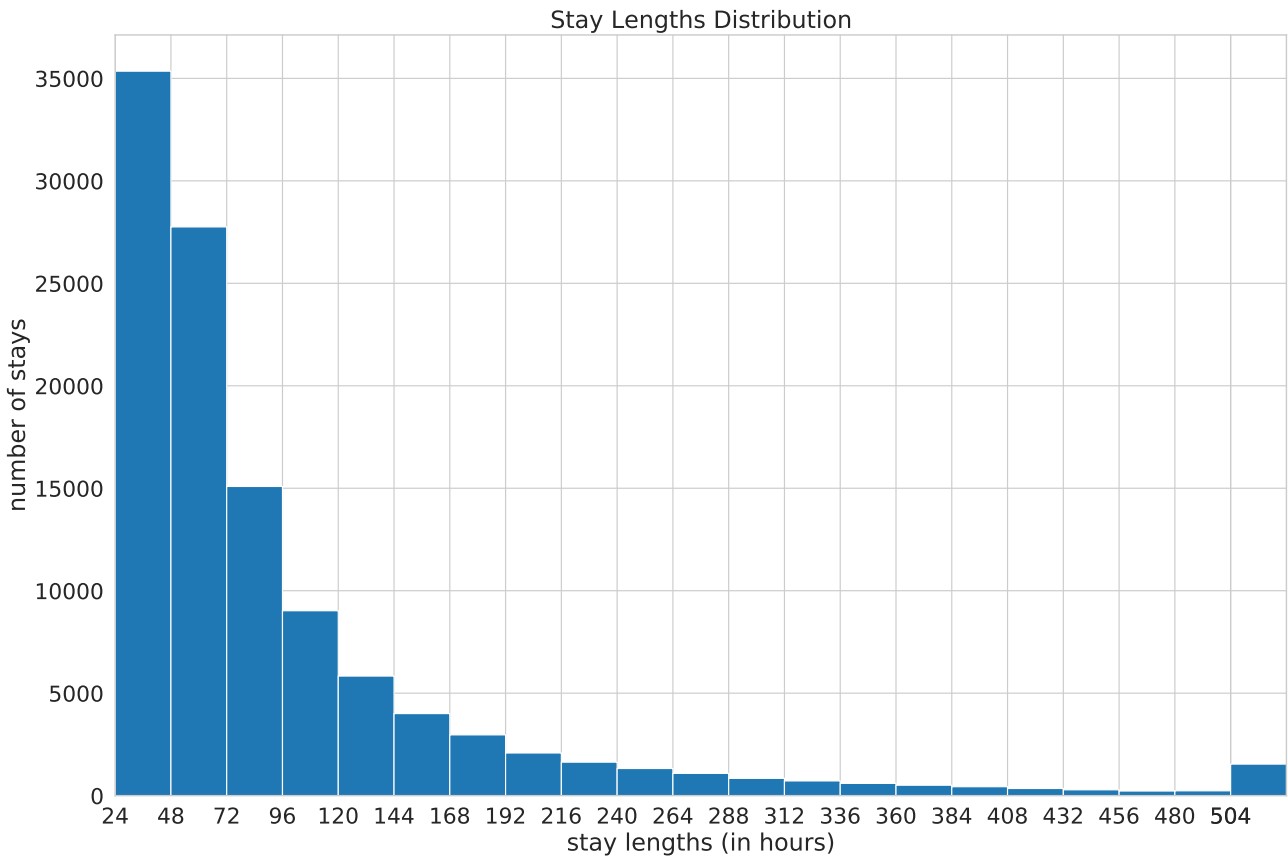

Figure E.2: Distribution of LOS in ICU obtained from eICU

## F. Model Training Time

| Dataset | | *seconds/epoch* to train each model | | | | |
|---|---|---|---|---|---|---|
| | | PC-HMM | GRU-D | MixMatch | FixMatch | BRITS |
| MIMIC-IV | 41 features | 9 | 43 | 70 | 55 | 48 |
| eICU | 35 features | 13 | 68 | 109 | 85 | 68 |

Table F.1: Computation time (seconds/epoch) required by each model. An epoch is completed when every example in the training set is covered at-least once. Although computation time increases when measurements are more frequent, the PC-HMM still requires far lesser time to train due to significantly lesser parameters than the other models.

## G. Baseline Semi-supervised learning adapted to Time-Series

Recent progress in improving deep classifiers in the SSL setting has been made under the family of *consistency regularization*. Following the unified analysis in Zhu et al. (2022), these approaches train a deep network $f_\theta$ with weights $\theta$ to minimize a two-term loss:

$$\sum_{X,y\in a(\mathcal{D}^L)} \ell(y, f_\theta(X)) + \lambda \sum_{X\in a(\mathcal{D}^U)} \ell(y'(X), f_\theta(X)) \tag{11}$$

Here, $\ell(\cdot,\cdot)$ is a loss function, $\lambda > 0$ is a tradeoff hyperparameter, $y'(X)$ is a labeling transformation, and $a(\cdot)$ represents an

(optional) data augmentation transformation. Below, we describe how both PseudoLabel and MixMatch fit this framework via concrete realizations of $y'$, $a$, and $\ell$.

In practice, to minimize this loss via minibatch gradient descent we start $\lambda$ at zero for a few hundred epochs, and gradually ramp up to a small positive value over a few hundred more epochs. This ensures that early learning fits well to the labeled set, while letting the unlabeled set also influence results later on.

**MixMatch for time series.** MixMatch (Berthelot et al., 2019) is a recent state-of-the-art SSL algorithm based on two key ideas. First, smooth transitions between classes in feature space are desirable and achievable via an interpolating augmentation scheme known as MixUp (Zhang et al., 2017). Second, it is useful to ensure consistency in the predicted label across multiple augmentations of the same source features. Originally designed for images, it has recently been applied to time series (Goschenhofer et al., 2021).

We set the labeling function $y'(X)$ to produce temperature-sharpened probability vectors averaged across multiple augmentations of examples X. The augmentation transformation $a(\cdot)$ uses MixUp to interpolate between labeled examples (see (Berthelot et al., 2019) for details). As a basic augmentation procedure applicable to time series, we add Gaussian noise $\mathcal{N}(0, \epsilon^2)$, following Goschenhofer et al. (2021), with standard deviation $\epsilon$ set to 0.1 and 1. For the backbone architecture $f_\theta$, we use a Gated Recurrent Unit(GRU) (Cho et al., 2014).

**FixMatch for time series.** Similar to MixMatch, FixMatch (Sohn et al., 2020) is another recent state-of-the-art SSL algorithm based on consistency regularization and pseudo-labeling. Pseudo-labels for weakly augmented unlabeled examples are only retained if the model produces high-confidence predictions (we try 0.6 and 0.8 as thresholds for high confidence predictions). Consistency regularization and the augmentation procedure is the same as MixMatch. Again, we use GRU for the backbone architecture.

