# OpenReview forum: "Semi-supervised Ordinal Regression via Cumulative Link Models for Predicting In-Hospital Length-of-Stay"
_ICML.cc/2023/Workshop/IMLH — IMLH 2023 PosterShortPaper_

### Official Review · Reviewer_FZXt · 2023-06-12
**A new method for predicting in-hospital length-of-stay**

**Rating:** 6
**Confidence:** 3

**Review:**

Summary: This short paper proposes using a cumulative link model to estimate the likelihood for ordinal regression. The method is evaluated on an in-hospital length-of-stay (LOS) task. It also combines this cumulative link model with a prediction-constrained hidden markov model for semi-supervised learning.

Novelty: This work applies the cumulative link model for ordinal regression on predicting length-of-stay (LOS). The application is novel and important, but the novelty of the method part is limited.

Soundness: This paper presents a comprehensive experimental study to show that using the cumulative link model can achieve comparable or better performance compared with SOTA. I have the following two concerns:
- Although the number of parameters is less, the gains of the performance with their methods are relatively small. In a real-world application, is there any other significant advantage of using this method?
- It would be helpful to add a comparison with multi-class classifiers and other regression models.

Clarity/Quality of writing: The paper is well-written. However, some important details are missing or are put in the appendix.
- Need more explanations and details of semi-supervised extensions.
- The last part of the experimentation section (section 4) demonstrates the interpretability of their method. But it lacks context and explanation to understand the figure.

Strength:
1. Length-of-stay is an important healthcare application. The topic is relevant to the theme of the workshop.
2. Cumulative link model is a parameter-efficient method for predicting LOS. It also considers a semi-supervised learning extension to help overcome unlabeled dataset challenges in the real-world.

Weakness:
1. It would be helpful to reorganize the structure of the paper. Many details are omitted in the main content now.
2. The interpretability part needs more explanation and elaboration.
3. The improvements of utilizing this cumulative link model shown in Table 2 are marginal compared to the previous binary models. Considering that previous work already utilized cumulative link models for ordinal data regression, the significance of the work is limited.

[1] Christensen, R. H. B. Cumulative link models for ordinal regression with the R package ordinal, 2022.

---

### Official Review · Reviewer_iw6r · 2023-06-15
**The idea is novel and the results are promising. But some experiments are missing.**

**Rating:** 6
**Confidence:** 5

**Review:**

Pros:
1.	The writing is good and the results are clear.
2.	The idea of using CLM is novel and its effectiveness is promising.
Cons:
1.	The authors mentioned the proposed method requires fewer parameters, but the efficiency comparison is missing in the manuscript.
2.	Why the performance of CLM in LOS > 3 is generally worse than the performance of binary classification?
3.	Could the authors provide the experiments based on a semi-supervised training setting?

---

### Official Review · Reviewer_s581 · 2023-06-16
**The paper is well-written while the method is not novel.**

**Rating:** 7
**Confidence:** 4

**Review:**

+ The paper is well-written with clear logic.
+ Results are strong.
+ Figures are designed well.

- PC-HMM is not a novel method.

---

### Official Review · Reviewer_gd4d · 2023-06-17
**Good paper, should be accepted**

**Rating:** 8
**Confidence:** 4

**Review:**

The authors observe that framing ordinal classification/regression tasks as multiple binary classifications for each outcome/bin of outcomes is crude, and propose that using cumulative link models can remedy this.They apply this to length of stay prediction and observe slightly better results than the binary approach. The authors present well-designed experiments that are described carefully.

I think this is very interesting work and should be accepted at the workshop.

There's some weaknesses/comments that the authors might wish to address in future iterations:
- You say: "This implementation of separate binary classi-fiers is not able to constrain a monotonically decreasing probability prediction from shorter to longer LOS classes (e.g. probability of LOS > 7 can be greater than that of LOS". I think that's generally fair, but not necessarily true. It could be that based on available info, it could be plausible that the patient either stays briefly or very long, but not a medium time; though these might be rarer cases.
- Despite your motivating observation, you evaluate the models as binary predictions. I think more careful examination of confusion-matrices (or a non-thresholded equivalent) would be in order in general for this task, but especially in your setting.
- The improvements are minor and could be due to parameter tying alone. Seeing improvements demonstrated with repeated CV and stronger baselines (e.g. MLP, but also tree-based models) would be interesting.

---

### Meta-Review · Area_Chair_Vx6K · 2023-06-18

**Recommendation:** Accept (Poster)
**Confidence:** 4

**Metareview:**

This paper proposed a cumulative link model to estimate the likelihood for ordinal regression. The method was evaluated on an in-hospital length-of-stay (LOS) task. It also combined this cumulative link model with a prediction-constrained hidden markov model for semi-supervised learning.

All reviewers found the paper interesting and sound, and recommended acceptance.

---

### Decision · Program_Chairs · 2023-06-20

Accept (Poster Short Paper)